# Resin Acid Copper Salt, an Interesting Chemical Pesticide, Controls Rice Bacterial Leaf Blight by Regulating Bacterial Biofilm, Motility, and Extracellular Enzymes

**DOI:** 10.3390/molecules29184297

**Published:** 2024-09-11

**Authors:** Lihong Shi, Xiang Zhou, Puying Qi

**Affiliations:** 1Guizhou Province Engineering Research Center of Medical Resourceful Healthcare Products, College of Pharmacy, Guiyang Healthcare Vocational University, Guiyang 550081, China; shilihong@zju.edu.cn; 2National Key Laboratory of Green Pesticide, Center for R&D of Fine Chemicals, Guizhou University, Guiyang 550025, China

**Keywords:** resin acid copper salt, chemical pesticide, anti-*Xoo* virulence, *gumB* gene, rice bacterial leaf blight

## Abstract

Bacterial virulence plays an important role in infection. Antibacterial virulence factors are effective for preventing crop bacterial diseases. Resin acid copper salt as an effective inhibitor exhibited excellent anti-*Xanthomonas oryzae* pv. *oryzae* (*Xoo*) activity with an EC_50_ of 50.0 μg mL^−1^. Resin acid copper salt (RACS) can reduce extracellular polysaccharides’ (EPS’s) biosynthesis by down-regulating *gumB* relative expression. RACS can also effectively inhibit the bio-mass of *Xoo* biofilm. It can reduce the activity of *Xoo* extracellular amylase at a concentration of 100 μg mL^−1^. Meanwhile, the results of virtual computing suggested that RACS is an enzyme inhibitor. RACS displayed good curative activity with a control effect of 38.5%. Furthermore, the result of the phytotoxicity assessment revealed that RACS exhibited slight toxicity compared with the control at a concentration of 200 μg mL^−1^. The curative effect was increased to 45.0% using an additional antimicrobial agent like orange peel essential oil. RACS markedly inhibited bacterial pathogenicity at a concentration of 100 μg mL^−1^ in vivo.

## 1. Introduction

Feng and Romanazzi et al. found that copper-based pesticides could prevent plant bacterial diseases effectively [1,2]. Dong et al. found that copper hydroxide could control Chinese cabbage black rot caused by *Xanthomonas campestris* pv. *campestris* [3]. Cazorla et al. found that Bordeaux mixture consistently decreased mango bacterial apical necrosis symptoms caused by *Pseudomonas syringae* pv. *syringae* [4,5] and controlled cherry bacterial canker disease [6]. Benmamoun and Carnegie et al. found that cuprous oxide not only controlled dothistroma needle blight resulting from *dothistroma septosporum* but also resulted in a reduction in viability for *Staphylococcus aureus*, *Pseudomonas aeruginosa*, and *Streptococcus pneumoniae* [7,8]. Lee and Duin et al. found that copper oxychloride prevented tomato wilt resulting from *Fusarium* [9], fruiter bacterial canker resulting from *Pseudomonas syringae* pv. *syringae* [10], and maize bacterial leaf streak resulting from *Xanthomonas vasicola* pv. *vasculorum* [11]. Cota-Ruiz et al. found that copper-based inorganic pesticides exhibited excellent antibacterial properties. Spraying copper-based inorganic pesticides, however, resulted in phytotoxicity, reduced root elongation, and inhibited the growth of roots [12]. Lu et al. found that frequently spraying Bordeaux mixture resulted in phytotoxicity to citrus during the management of citrus scab and citrus canker, increased leaf Cu content, destroyed the ultrastructural integrality of chloroplasts, decreased photosynthesis, and resulted in leaf chlorosis [13].

However, copper-based organic pesticides, such as thiodiazole copper, thiosen copper, and resin acid copper salt (shown in Figure 1), are more secure than copper-based inorganic pesticides and do not cause phytotoxicity for target crops due to their lower copper content and lower accumulation of copper. Chen and Dai et al. found that thiodiazole copper and thiosen copper could control rice bacterial leaf blight [14,15,16], kiwifruit bacterial canker [17,18], and citrus bacterial canker [19,20]. Zhang and Madhu Sekhar et al. found that copper-based bactericides containing 1, 3, 4−thiadiazole fragments could efficiently prevent crop bacterial diseases attributed to cell membrane disruption [21], cytoderm disruption, and copper-induced cell death [17,22].

In addition, RACS, another important commercial pesticide, exhibits excellent control efficacy for citrus bacterial canker [23]. Meanwhile, studies on RACS preventing rice bacterial leaf blight and its antibacterial mechanisms are rarely reported. Favaro et al. found that copper ions not only destroyed cell membranes and cytoderm but also inhibited the formation of bacterial biofilm [24]. Ito and Dzięgielewska et al. found that resin acid not only inhibited the growth and biofilm formation of *Streptococcus mutans* [25] but also disturbed the biofilms’ biosynthesis of *Staphylococcus aureus*, *Pseudomonas aeruginosa*, and *Candida albicans* [26]. Resin acid amino alcohol derivatives based on the resin acid skeleton show excellent anti-virulence bioactivity against *Xoo*. In this work, we studied the bioactivity of RACS against *Xoo* in vitro and in vivo, and its potential anti-virulence mechanism was further investigated.

## 2. Results and Discussion

### 2.1. Anti-Xoo Potency Assessment of RACS

Table 1 displays the anti-*Xoo* potency of RACS. The inhibitory rates were 100 ± 2.31, 64.3 ± 1.52, and 50.2 ± 1.63 at doses of 500, 100, and 50 μg mL^−1^, respectively. The value of EC_50_ was 50.0 ± 4.43 μg mL^−1^. The regression equation was y = 1.15x + 3.06, and the R square value was 0.9996. Based on this data analysis, RACS exhibited significant anti-*Xoo* activity. Previous work revealed that RACS exhibited excellent control efficacy for citrus bacterial canker caused by *Xanthomonas axonopodis* pv. *citri* (*Xac*) [23]. *Xoo* was also a *Xanthomonas* as *Xac*. We boldly speculate that RACS also displayed anti-*Xoo* activity as *Xac* from these findings.

### 2.2. The Analysis of RACS against Xoo Biofilm Activity

Biofilm is an important virulence factor consisting of bacterial communities and is involved in bacterial quorum sensing [27]. A total of 65–80% of bacterial infections are related to biofilm [28]. Biofilms, physical barriers, have adapted to all kinds of hostile environmental changes [29] and increased the tolerance to adapt the hosts and antibiotic therapy [30]. EPSs are the major components of biofilm [31], and they play important roles in the bacterial pathogenic process just like biofilm [32,33]. Extracellular polysaccharides can facilitate bacteria to adhere and colonize [34]. *Gum* family genes regulate the biosynthesis of *Xoo* EPS [35,36]. Thereby, the anti-biofilm activity of RACS was measured by the crystal violet staining method [37]. The results revealed that RACS exhibited a strong ability to suppress the formation of biofilm at the concentrations of the control value (0), 12.5, 25.0, 50.0, and 100 μg mL^−1^. This effect was enhanced with the rise in the concentration of RACS. As shown in Figure 2, the inhibitory rates of biofilm formation were 0, 15.0 ± 1.94%, 30.6 ± 1.24%, 53.4 ± 4.48%, and 64.7 ± 2.83% when the concentrations were the control value, 12.5, 25.0, 50.0, and 100 μg mL^−1^. Furthermore, the results of RT-qPCR showed that the relative expressions of the *gumB* gene were 0.357 and 0.184 with doses of 50.0 and 100 μg mL^−1^. The potential antibacterial mechanism might be due to RACS disturbing the biosynthesis of EPSs by regulating the *gumB* gene and further reducing biofilm production.

### 2.3. The Analysis of Xoo Biofilm Formation Using Acridine Orange Staining

Biofilm formation contributed to the promotion of bacterial invasion and infection [38,39]. The acridine orange staining method was used to analyze the biofilm formation quantitatively [40]. As shown in Figure 3, these findings displayed that RACS could markedly suppress *Xoo* biofilm formation, and this inhibition effect was elevated gradually as the concentrations of RACS increased. The inhibitory rates were 23.08 ± 7.28%, 41.46 ± 1.16%, 49.83 ± 1.61%, and 69.05 ± 7.70% with concentrations of 12.5, 25.0, 50.0, and 100 μg mL^−1^. These outcomes were analogous to crystal violet staining. Based on these findings, RACS could significantly suppress bacterial virulence by regulating the *gumB* gene and biofilm’s biosynthesis. Thereby, RACS might be regarded as a virulence inhibitor for controlling plant disease.

### 2.4. Inhibited Effects on Xoo Motility

The majority of bacteria exhibited outbound motility [41,42], which enabled the microbial community to perceive environmental alterations [43,44,45] and pursue good fortune and avoid disaster [46]. Microbial motility played an important role in nutrients. It enabled bacteria to colonize complicated environments and adhere to the host surface to form biofilms [47], which contributed to enhancing bacterial adaption [48]. In addition, motility was closely associated with bacterial virulence [49]. As shown in Figure 4, bacterial motility diameter was increased with the increase in dose, and the diameters at doses of the control value (0), 12.5, 25.0, 50.0, and 100 μg mL^−1^ were 31 ± 1.0, 27 ± 1.7, 24 ± 1.7, 22 ± 2.3, and 20 ± 1.5 mm. The inhibitory ratios of motility corresponding to diameter were 0, 12.90 ± 5.88%, 22.58 ± 5.88%, 27.96 ± 7.45%, and 36.56 ± 4.93%. Therefore, *Xoo* motility was reduced markedly under the influence of RACS. This may be because the antibacterial activity was enhanced by the presence of copper ions and resin acid. Copper ions and resin acid could disturb the formation of biofilm. Meanwhile, the reduction in *Xoo* motility might result in the decline in bacterial virulence, which enabled bacteria to not adapt to environmental changes, effectively decreased bacterial infections, and obviously reduced the morbidity of plant diseases. So, RACS might be an inhibitor of *Xoo* motility.

### 2.5. The Inhibition Effect of Xoo Extracellular Hydrolases

The pathogenic microorganisms secreted numerous extracellular hydrolases [50,51,52] to hydrolyze host cell walls, destroying cellulose, starch, pectin, and xylan [53,54,55,56], which destroyed cell homeostasis, facilitating bacterial invasion and infection, and elevated the disease process. So, extracellular hydrolases were important virulence factors [57,58,59]. Destroying extracellular hydrolases would contribute to the remission of infection. As shown in Figure 5A,B, RACS significantly inhibited the activity of extracellular amylase, and the inhibitory ratio was 33.75 ± 7.21% at a dose of 100 μg mL^−1^. As shown in Figure 5C, we further used virtual computing (http://www.molinspiration.com, accessed on 3 August 2024) to validate the results of RACS as an enzyme inhibitor. The score of virtual computing was 0.13, and this meant that RACS was an enzyme inhibitor.

### 2.6. The Control Effect and Phytotoxicity Evaluation of RACS against Xoo at a Dose of 200 μg mL^−1^ In Vivo

RACS exhibited excellent bioactivity against *Xoo* in vitro and an attractive anti-virulence mechanism. Effectively controlling rice bacterial leaf blight was our final purpose. As shown in Figure 6 and Table 2, RACS displayed good curative activity with a control effect of 38.5%. Furthermore, the result of the phytotoxicity evaluation suggested that RACS displayed minor toxicity compared with the control.

### 2.7. The Analysis of Absorption, Distribution, Metabolism, and Excretion Using the Software of ADMETlab 2.0

Inadequate pharmacokinetics and drug toxicity evaluations of pesticide candidates are some of the important reasons for the late-stage failure of pesticide discovery. The evaluation of the absorption, distribution, metabolism, excretion, and toxicity (ADMET) was widely recognized using the software of ADMETlab 2.0 (https://admetmesh.scbdd.com, accessed on 23 July 2024). As shown in Figure 7, the ADMET Evaluation function module is composed of a series of high-quality prediction models trained by a multi-task graph attention framework. It enables the users to conveniently and efficiently implement the calculation and prediction of 17 physicochemical properties, 13 medicinal chemistry measures, 23 ADME endpoints, 27 toxicity endpoints, and 8 toxicophore rules (751 substructures), thereby selecting promising lead compounds for further exploration (https://admetmesh.scbdd.com, accessed on 23 July 2024). As shown in Figure 8, the results suggested that RACS exhibited pharmacological and toxicological features. The results of SAscore, Fsp3, MCE-18, Caco-2 permeability, MDCK permeability, Pgp-substrate, HIA, VD, BBB Penetration, DILI, AMES Toxicity, Carcinogenicity, Eye corrosion, NR-AhR, and NR-PPAR-gamma were fine. Thereby, RACS exhibited some potential in the ADMET Evaluation. It might be used as a bactericide for controlling rice bacterial leaf blight in the future.

### 2.8. The Control Effect Evaluation of RACS Containing 0.3% Orange Peel Essential Oil against Xoo at a Dose of 200 μg mL^−1^ In Vivo

Although the curative effect of RACS is evaluated in Figure 6, the effect was undesirable. So, we hoped to discover an effective pesticide adjuvant for increasing the curative activity of RACS. Based on previous studies [60], orange peel essential oil (OPEO) was an effective pesticide adjuvant derived from plant essential oil with wide application. As shown in Figure 9 and Table 3, the contact angle in the curative effect evaluation of RACS containing 0.3% orange peel essential oil against *Xoo* was 45.0% at a dose of 200 μg mL^−1^ in vivo. Additionally, the contact angle of RACS containing 0.3% OPEO was 56 degrees. The contact angles of water and RACS (20%, *m*/*v*, emulsion in water) were 119 and 59 degrees. The results revealed that pesticide adjuvant increased the control effect by 6.5% compared with RACS due to the reduction in contact angle. It was very exciting.

### 2.9. Influence of RACS on Bacterial Pathogenicity on Rice Plant at Control and 100 μg/mL

Although RACS displayed excellent anti-virulence activity for controlling rice bacterial leaf blight, the pathogenicity must be verified in vivo. The influence of RACS on bacterial pathogenicity was evaluated at concentrations of the control value (0 μg/mL) and 100 μg/mL. As shown in Figure 10, RACS markedly inhibited the occurrence of rice bacterial disease and lesions with a length of 11.1 ± 4.9 cm and 1.1 ± 0.6 cm at the concentrations of the control value and 100 μg/mL. Therefore, RACS markedly inhibited bacterial pathogenicity.

## 3. Materials and Methods

### 3.1. Instruments and Chemicals

Sterile polystyrene 6-well plates and 96-well plates were purchased from Shanghai Titan Technology Co., Ltd., Shanghai, China. Crystal violet (purity, 98%), dimethylsulfoxide (DMSO, purity, 98%), and tetramethylacridine-3,6-diamine hydrochloride (purity, 98%) were obtained from Shanghai Bide Pharmaceutical Technology Co., Ltd., Shanghai, China. Ammonium oxalate monohydrate (purity, 98%), iodine (purity, 99.8%), and starch soluble (purity, AR) were obtained from Saen Chemical Technology Co., Ltd., Shanghai, China. The TaqMan RT-qPCR Kit (Code, NO. RR037A) was bought from Beijing Solarbio Science & Technology Co., Ltd. (Beijing, China). Orange peel essential oil was purchased from Henan Moore Water soluble Fertilizer Co., Ltd. (No.11, Unit 4, Building 9, Hongnan Road Electronic and Electrical Industrial Park, Zhengzhou High-tech Industrial Development Zone, Zhengzhou city, Henan Province, China). Resin acid copper salt (20%, water emulsion) was purchased from Qingdao Haina Bio-Technology Co., Ltd., Qingdao, China. The analysis of bacterial fluorescence imaging was conducted using the instrument of the Olympus-BX53 microsystem (Olympus Corporation, Tokyo, Japan). The HARKE SPCAX3 contact angle instrument was from Beijing, China.

### 3.2. The Potency Assessment of RACS against Xoo and Biofilm

The potency assessment of RACS against *Xoo* and biofilm was carried out based on previous works [61].

### 3.3. Relative Expression Analysis of gumB Gene Coding EPS Using RT-qPCR

The relative expression analysis of the *gumB* gene coding EPSs was carried out by using the TaqMan RT-qPCR Kit (Code, NO. RR037A) [62]. Firstly, 40 μL *Xoo* suspension (OD_595_ = 0.6) was added to a 15 mL glass test tube with 5 mL LB medium containing RACS (control, 50, and 100 μg mL^−1^), and the mixture was cultured for 72 h. Secondly, the *Xoo* precipitation was extracted at the bottom of the tube by 7000 rpm centrifugation, and the mRNA was further obtained from the *Xoo* precipitation using the TransZol UP kit. Subsequently, the relative expression of the *gumB* gene was measured using the mRNA and RT-qPCR Kit. Finally, the data were analyzed using the CFX96 PCR System (Bio-Rad, Hercules city, CA, USA).

### 3.4. Fluorescent Staining Assay of Xoo Biofilm

The OD_595_ of 4 mL LB bacterial medium with *Xoo* suspension was adjusted to 0.1. Then, LB bacterial medium was added to the sterile polystyrene 6-well plate. Meanwhile, RACS at doses of the control value (0), 12.5, 25.0, 50.0, and 100 μg mL^−1^ was added to a 6-well plate. Subsequently, a glass sheet (10 mm × 10 mm) was placed in every well. These 6-well plates were sealed and cultured in a constant temperature incubator at 28 °C for 72 h. Next, each glass sheet was stained using acridine orange dye [63]. Finally, the data were analyzed using software and an Olympus-BX53 microsystem.

### 3.5. Motility Assessment of RACS against Xoo

The heated 4 mL semisolid medium including 0.5% agar was poured into a 6-well plate. Then, various doses of RACS were added to each well to form the mixture with different concentrations of the control value (0), 12.5, 25.0, 50.0, and 100 μg mL^−1^. Next, 10 μL *Xoo* cell suspension (OD_595_ = 0.6) was added to the plate’s center after cooling the semisolid medium. A 6-well plate including medium, *Xoo*, and RACS was placed and co-cultured in a constant temperature incubator at 28 °C for 72 h. Lastly, the diameter of motility was observed and tested [64].

### 3.6. The Analysis of Xoo Extracellular Hydrolase Activity

The heated 4 mL semisolid medium including 0.5% agar and 0.1% soluble starch was poured into a 6-well plate. Then, various doses of RACS were added to each well to form the mixture with different concentrations of the control value (0), 12.5, 25.0, 50.0, and 100 μg mL^−1^. Next, 10 μL *Xoo* cell suspension (OD_595_ = 0.6) was added to the plate’s center after cooling the semisolid medium. A 6-well plate including medium, *Xoo*, and RACS was placed and co-cultured in a constant temperature incubator at 28 °C for 72 h. Lastly, the medium was stained with iodine/potash iodide solution and washed using 70% ethanol solution. The diameter of the motility plate was observed and tested [64].

### 3.7. The Curative Activity and Influence of RACS on Bacterial Pathogenicity against Xoo In Vivo

RACS solution at doses of the control value (0) and 200 μg mL^−1^ was prepared. Rice at the tillering stage was obtained in a phytotron. *Xoo* cell suspension (OD_595_ = 0.6) was inoculated into leaves by cutting leaves [64]. Then, RACS solution was sprayed onto leaves after 24 h. Finally, these rice plants were cultured in a phytotron for 14 days, and all leaf lesion lengths were tested and calculated.

### 3.8. The Phytotoxicity Evaluation of RACS In Vivo

RACS solution at doses of the control value (0) and 200 μg mL^−1^ was prepared. Rice at the tillering stage was obtained in a phytotron. Then, RACS solution was sprayed onto the leaves. Finally, these rice plants were cultured in a phytotron for 14 days, and all leaf lesion lengths were tested and calculated [64].

### 3.9. The Analysis of Absorption, Distribution, Metabolism, Excretion, and Toxicity (ADMET)

The analysis of ADMET was carried out using the software of ADMETlab 2.0 (https://admetmesh.scbdd.com, accessed on 23 July 2024). Firstly, the SMILES of RACS were copied into “ADMET Evaluation-SMILES” and submitted. Secondly, the results were exhibited in a PDF. Finally, a green filled circle indicates good results, a red filled circle indicates bad results, and a yellow filled circle indicates general results [65].

### 3.10. Statistical Analysis

All experiments were conducted with three replicates. An analysis of variance (ANOVA) was used to evaluate the differences between each experiment (Origin 2021, OriginLab Company, Hampton, MA, USA). Values (a, b, and c) in a panel with different letters are markedly different (*p* < 0.05) according to Turkey’s multiple comparison post-test.

## 4. Conclusions

In summary, resin acid copper salt, an important commercial pesticide, exhibits excellent control efficacy for citrus bacterial canker caused by *Xanthomonas axonopodis* pv. *citri*. Meanwhile, studies on RACS controlling rice bacterial leaf blight and its antibacterial mechanism are rarely reported. In this work, we studied in detail the antibacterial mechanism and control efficacy of RACS for rice bacterial leaf blight. The results suggested that RACS exhibited good curative activity (38.5–45.0%) and excellent anti-virulence function. Firstly, RACS prevented EPSs’s biosynthesis by adjusting *gumB* gene expression and further decreased the production of biofilm. Secondly, RACS reduced bacterial virulence by regulating the motility and the activity of extracellular amylase. Finally, RACS significantly reduced bacterial pathogenicity.

## Figures and Tables

**Figure 1 molecules-29-04297-f001:**
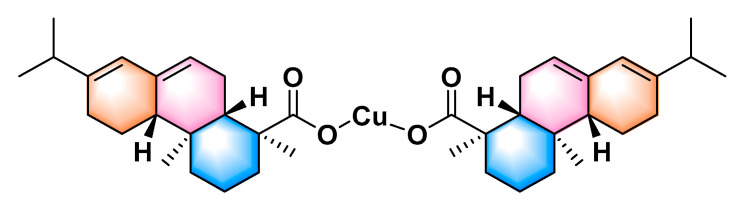
The chemical structure of RACS.

**Figure 2 molecules-29-04297-f002:**
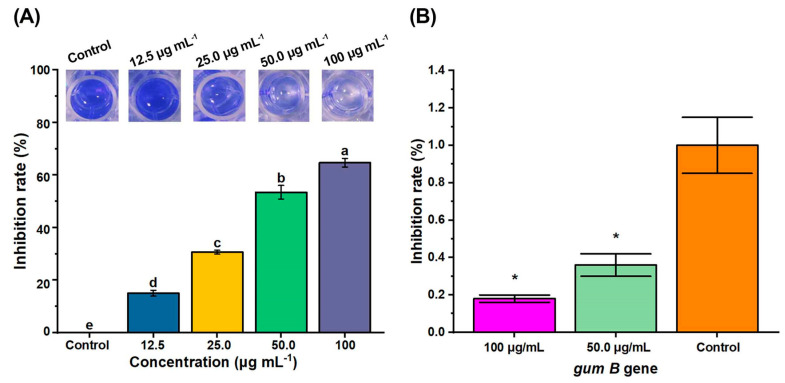
The quantitative assay of crystal violet staining *Xoo* biofilm. (**A**) The inhibition rate of RACS regulated biofilm production at doses of the control value (0), 12.5, 25.0, 50.0, and 100 μg mL^−1^. (**B**) The relative normalized expression of the *gumB* gene at doses of the control value, 50.0, and 100 μg mL^−1^. Lowercase letters above the histogram indicate significant differences (*p* < 0.5). (* *p* < 0.05).

**Figure 3 molecules-29-04297-f003:**
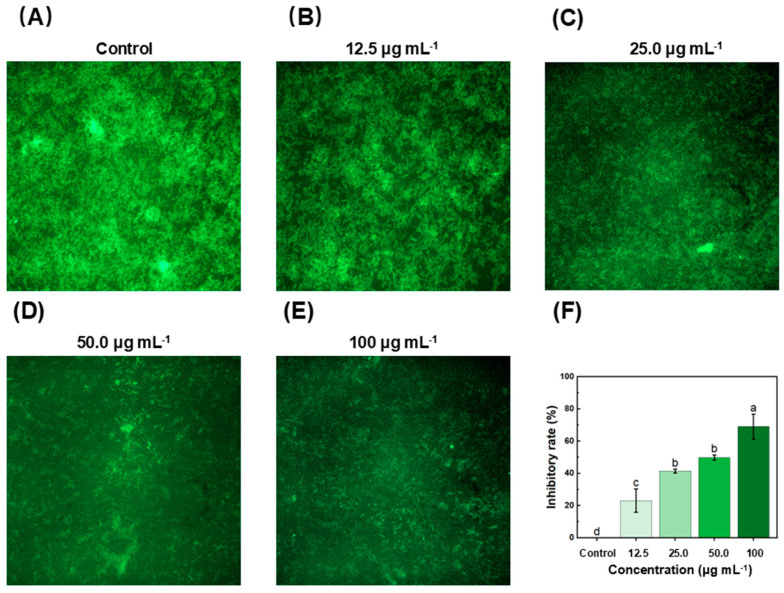
Analysis of Xoo biofilm formation using acridine orange staining. (**A**–**E**) Acridine orange staining *Xoo* biofilms. (**F**) Inhibitory rate of *Xoo* biofilm production. Lowercase letters above dots indicate significant differences (*p* < 0.5).

**Figure 4 molecules-29-04297-f004:**
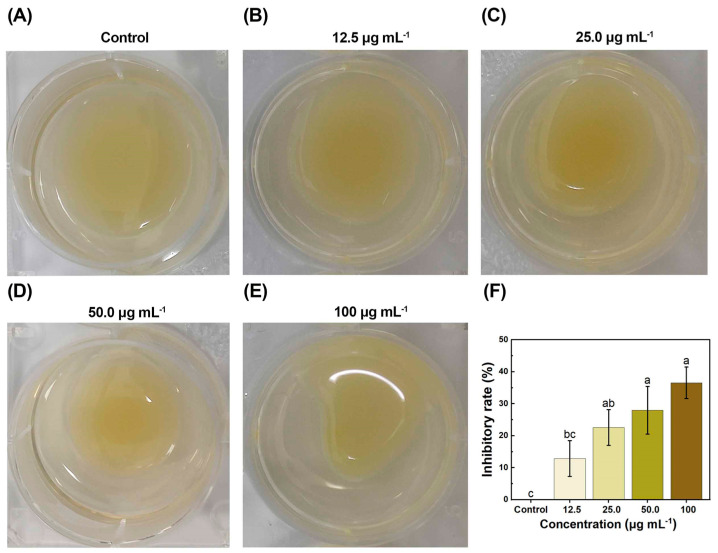
The analysis of *Xoo* motility. (**A**–**E**) RACS suppressed the *Xoo* motility at doses of the control value (0), 12.5, 25.0, 50.0, and 100 μg mL^−1^. (**F**) The inhibitory ratio of *Xoo* motility. The scale plate was 10 mm. Lowercase letters above the dots indicate significant differences (*p* < 0.5).

**Figure 5 molecules-29-04297-f005:**
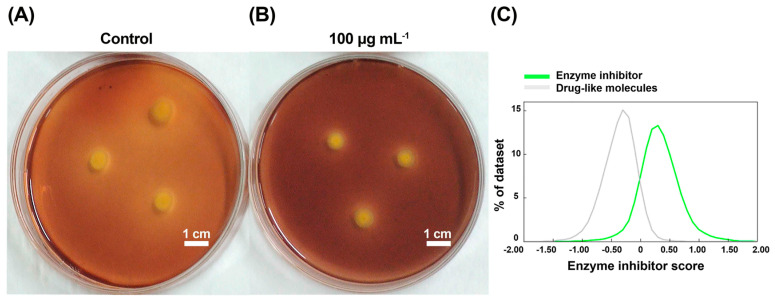
The activity of *Xo* extracellular amylase at concentrations of the control value (0) and 100 μg mL^−1^ (**A**,**B**). (**C**) The results of RACS as an enzyme inhibitor by virtual computing. Scale bar = 1 cm.

**Figure 6 molecules-29-04297-f006:**
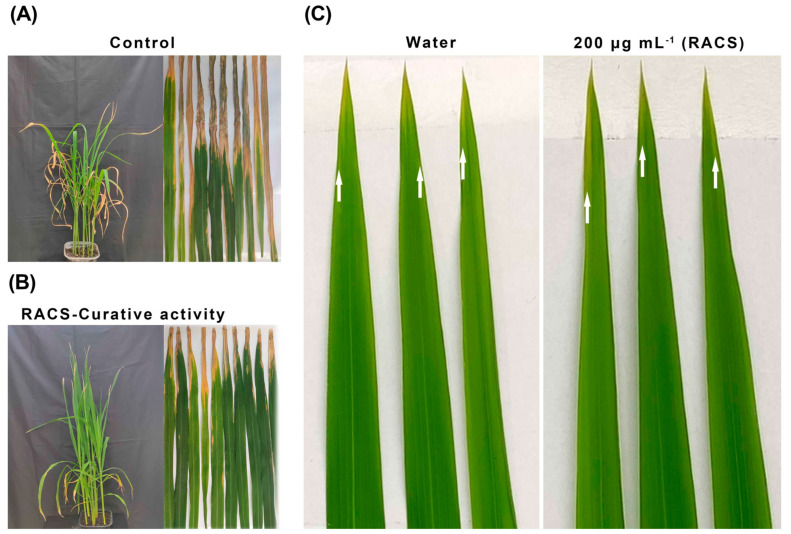
The control effect and phytotoxicity evaluation of RACS against *Xoo* at a dose of 200 μg mL^−1^ in vivo. (**A**) Control. (**B**) RACS curative activity. (**C**) Phytotoxicity evaluation. The white arrow indicates the lesion length.

**Figure 7 molecules-29-04297-f007:**
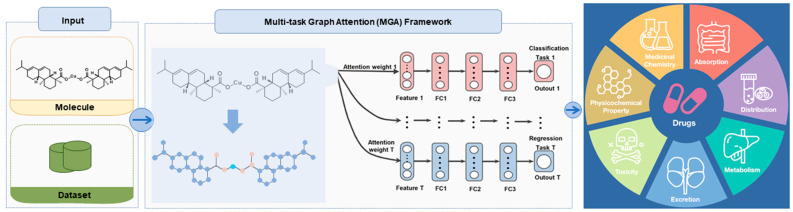
The multi-task graph attention (MGA) framework of the ADMET Evaluation.

**Figure 8 molecules-29-04297-f008:**
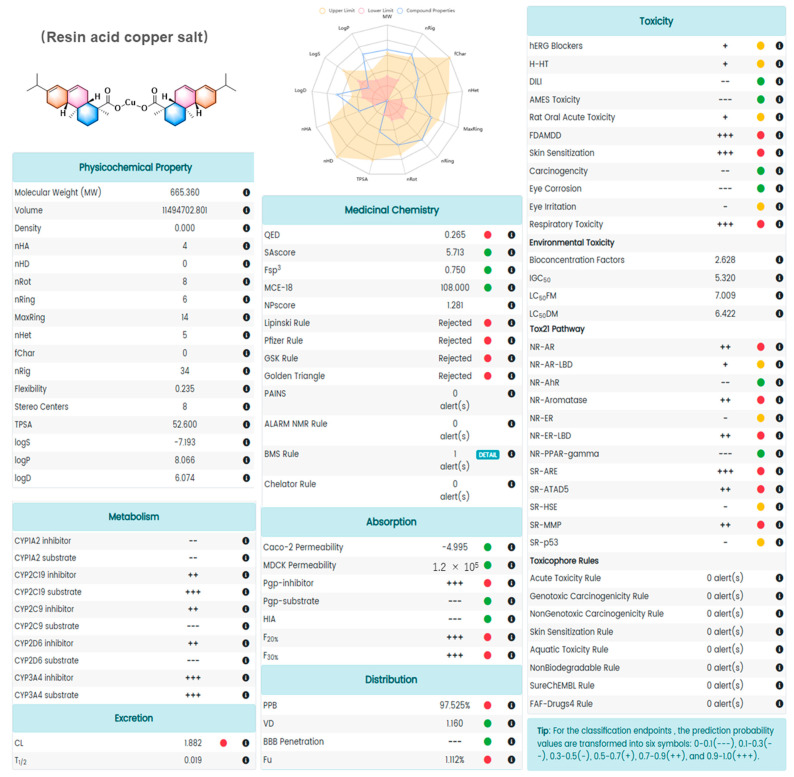
The results of the ADMET Evaluation. A green filled circle indicates good results, a red filled circle indicates bad results, and a yellow filled circle indicates general results.

**Figure 9 molecules-29-04297-f009:**
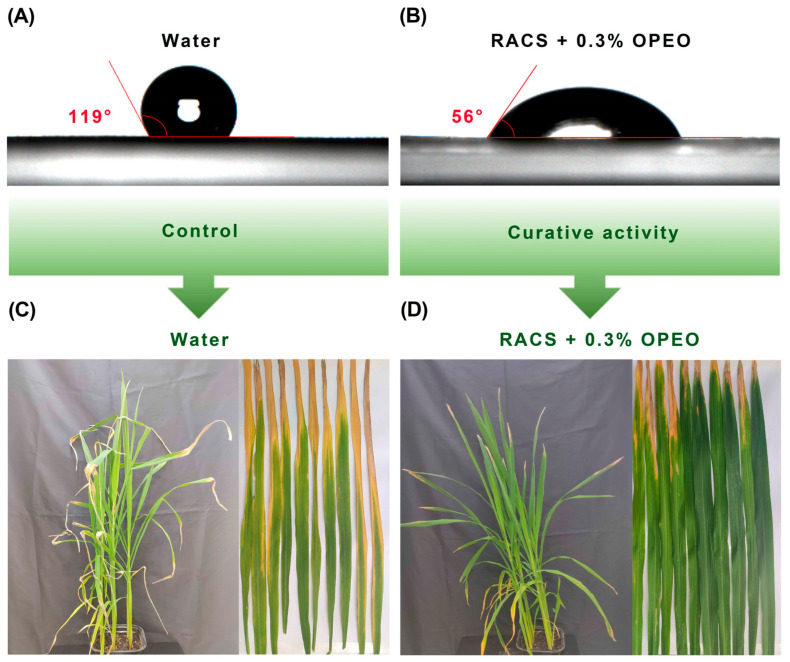
The curative effect evaluation of RACS containing 0.3% OPEO against *Xoo* at a dose of 200 μg mL^−1^ in vivo. (**A**) The contact angle of water. (**B**) The contact angle of RACS + 0.3% OPEO. (**C**) The curative effect evaluation of the control. (**D**) The curative effect evaluation of RACS + 0.3% OPEO.

**Figure 10 molecules-29-04297-f010:**
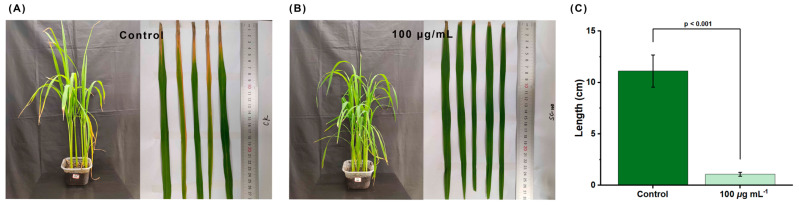
The influence of RACS on bacterial pathogenicity on rice plant at the control value (**A**) and 100 μg/mL (**B**). (**C**) The length of lesions.

**Table 1 molecules-29-04297-t001:** In vitro antibacterial potency of compounds of RACS against *Xoo*.

	*Xoo*					
Compound	500 (µg mL^−1^)	100 (µg mL^−1^)	50 (µg mL^−1^)	Regression Equation	EC_50_ (µg/mL) ^a^	R^2^
RACS	100 ± 2.31	64.3 ± 1.52	50.2 ± 1.63	y = 1.15x + 3.06	50.0 ± 4.43	0.9996

^a^ The EC_50_ values of antibacterial activities are indicated as the mean ± SD (standard error) for three independent repetitions.

**Table 2 molecules-29-04297-t002:** In vivo curative activity of RACS at dose of 200 μg mL^−1^.

Treatment	Curative Activity		
	Morbidity (%)	Disease Index (%)	Control Efficiency (%) ^b^
RACS	100	56.7	38.5
Control ^a^	100	92.2	/

^a^ Control. ^b^ Statistical analysis was performed using ANOVA under condition of equal variances assumed (*p* < 0.05).

**Table 3 molecules-29-04297-t003:** In vivo curative activity of RACS containing 0.3% OPEO at dose of 200 μg mL^−1^.

Treatment	Curative Activity		
	Morbidity (%)	Disease Index (%)	Control Efficiency (%) ^b^
RACS + 0.3% OPEO	100	51.2	45.0
Water ^a^	100	93.1	/

^a^ Negative control (water). ^b^ Statistical analysis was performed using ANOVA under condition of equal variances assumed (*p* < 0.05).

## Data Availability

The original contributions presented in this study are included in this article; further inquiries can be directed to the corresponding authors.

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
