# Peer review of "Resin Acid Copper Salt, an Interesting Chemical Pesticide, Controls Rice Bacterial Leaf Blight by Regulating Bacterial Biofilm, Motility, and Extracellular Enzymes"

_molecules, 2024, doi:10.3390/molecules29184297_

Round 1
Reviewer 1 Report
Comments and Suggestions for Authors
Dear authors
Happy day
The paper is fine but need some improvement.
Kindly consider my suggestions and comments.
The title
1- The words “regulating bacterial virulence” sound non-specific and give the impression about controlling all the virulence factors. Kindly be specific according to your finding.
In the abstract part
2- “anti-Xoo” kindly write the full scientific name Xanthomonas oryzae pv. oryzae.
3- RACS kindly write the full name before the abbreviation.
4- “Meanwhile, the results of virtual computing (in silico) suggested the RACS as an enzyme inhibitor.” Why you like to put that as a suggestion why not to do a simple experiment to show a real inhibition for the amylase using your compound. Kindly, explain carefully. Did you mean that you have meagered the activity using a software?! This also will be different from using the expression “in silico”
5- “pesticide adjuvant” this expressions has different definitions I suggest to remove it and simply to use a simple words like “additional antimicrobial agent like----” The scientific writing in its best format usually avoid using of sounded words or that need a dictionary or a definition (for non-specialists). Adjuvant is usually a term used by the immunologist. Even so, your compound did not lead to the production of a new plant product that lead to the killing of the target microbes. I suggest to rephrase the sentence.
6- “Finally” you might be able to use such word in the result section but it sound odd in the abstract because you did not list all your finding to say “finally”, kindly use a better word.
In the Introduction part
7- The first two sentences you describe the Copper-based pesticides but it is better to describe the authors whom prove/describe that. Better to do that in all the text.
In the Results and Discussion part
8- The images not good.
In the material and methods part
9- You have used the word “Firstly” in the beginning of different parts in the material and methods. Kindly use different words or just ignore it.
10- In 3.6. part “Finally, the good or bad information was analyzed” This phrase must be changed or deleted from the text. However perhaps the software categories the results as “good” and “bad”; so you need to describe that.
The figures parts
11- Figure 10 as an example: this is your ligand: “. The Analysis of Pathogenicity of RACS on Rice Plant at 0 and 100 μg mL-1”: Kindly : 1) where the control? Where the description of “C”? “mL-1” need a correction (remove the full stop)
References
12- Nothing to concern
General comment:
13- You must use a real enzyme activity test.
14- The motility test is not representative, kindly give better images, or just change the experiment.
15- Kindly in most of the images there is no control. Kindly add some images to the control.
16- You represent good images for the plants but you need the represent a correct killing to the microbes using your compound either by using a serial dilation method agar well diffusion experiment, etc.
With my pleasure

Author Response
The title
Comments 1: The words “regulating bacterial virulence” sound non-specific and give the impression about controlling all the virulence factors. Kindly be specific according to your finding.
Response 1: We thank the expert reviewer for the overall affirmative comment in regards of our manuscript. Thank you for your kind suggestions. Your suggestion will be of great help to our research. We already had revised the Title with your suggestion, and the new title is “Resin acid copper salt, an interesting chemical pesticide, controlling rice bacterial leaf blight by regulating bacterial biofilm, motility, and extracellular enzyme”.
In the abstract part
Comments 2: “anti-Xoo” kindly write the full scientific name Xanthomonas oryzae pv. oryzae.
Response 2: Thank you for your kind suggestions of our manuscript. We already had revised the “anti-Xoo” with your suggestion in the abstract.
Comments 3: RACS kindly write the full name before the abbreviation.
Response 3: Thank you for your kind suggestions of our manuscript. We already had supplied the abbreviation of RACS with your suggestion in the abstract.
Comments 4: “Meanwhile, the results of virtual computing (in silico) suggested the RACS as an enzyme inhibitor.” Why you like to put that as a suggestion why not to do a simple experiment to show a real inhibition for the amylase using your compound. Kindly, explain carefully. Did you mean that you have meagered the activity using a software?! This also will be different from using the expression “in silico”.
Response 4: Thank you for your kind suggestions of our manuscript. Your suggestion will be of great help to our research. We did not express clearly about this part. Firstly, we completed the experiment of extracellular amylase using iodine solution staining. Secondly, as shown in Figures 5A and 5B, RACS significantly inhibited the activity of extracellular amylase and the inhibitory ratio was 33.75±7.21% at the dose of 100 μg mL−1. Lastly, we predicted RACS as an enzyme inhibitor using virtual computing. The expression “in silico” would be deleted in the revised manuscript. Thank you for your kind suggestions again.
Comments 5: “pesticide adjuvant” this expressions has different definitions I suggest to remove it and simply to use a simple words like “additional antimicrobial agent like----” The scientific writing in its best format usually avoid using of sounded words or that need a dictionary or a definition (for non-specialists). Adjuvant is usually a term used by the immunologist. Even so, your compound did not lead to the production of a new plant product that lead to the killing of the target microbes. I suggest to rephrase the sentence.
Response 5: Thank you for your kind suggestions of our manuscript. We already had revised the “pesticide adjuvant” with your suggestion (additional antimicrobial agent like) in the abstract.
Comments 6: “Finally” you might be able to use such word in the result section but it sound odd in the abstract because you did not list all your finding to say “finally”, kindly use a better word.
Response 6: Thank you for your kind suggestions. Your suggestion will be of great help to our research. We already had removed the “Finally” in the revised manuscript.
In the Introduction part
Comments 7: The first two sentences you describe the Copper-based pesticides but it is better to describe the authors whom prove/describe that. Better to do that in all the text.
Response 7: Thank you for your kind suggestion. Your suggestion will be of great help to our research. We already had revised it in the revised manuscript. Such as, 1. Introduction: Feng and Romanazzi etal., found that copper-based pesticides could prevent plant bacterial diseases effectively [1, 2]. Dong etal., found that copper hydroxide could control Chinese cabbage black rot caused by Xanthomonas campestris pv. campestris [3]. Cazorla etal., found that bordeaux mixture consistently decreased mango bacterial apical necrosis symptoms caused by Pseudomonas syringae pv. syringae [4, 5] and controls cherry bacterial canker disease [6]. Benmamoun and Carnegie etal., found that cuprous oxide not only controled dothistroma needle blight resulting from dothistroma septosporum but also resulted in a reduction of viability for Staphylococcus aureus, Pseudomonas aeruginosa, and Streptococcus pneumoniae [7, 8]. Lee and Duin etal., found that copper oxychloride prevented tomato wilt resulting from Fusarium [9], fruiter bacterial canker resulted from Pseudomonas syringae pv. syringae [10], and maize bacterial leaf streak resulted from Xanthomonas vasicola pv. vasculorum [11]. Cota-Ruiz etal., found that copper-based inorganic pesticides exhibited excellent antibacterial properties. Spraying copper-based inorganic pesticides, however, resulted in phytotoxicity, reduced root elongation, and inhibited the growth of root [12]. Lu etal., found that frequently spraying bordeaux mixture resulted in phytotoxicity to citrus during the management of citrus scab and citrus canker, increased the leaf Cu content, destroys the ultrastructural integrality of chloroplast, decreased the photosynthesis, and resulted in leaf chlorosis [13].
However, copper-based organic pesticides, such as thiodiazole copper, thiosen copper, and resin acid copper salt (Shown in Figure 1), are more secure than copper-based inorganic pesticides and it not cause phytotoxicity for target crops due to their lower copper content and less accumulation of copper. Chen and Dai etal., found that thiodiazole copper and thiosen copper could control rice bacterial leaf blight [14-16], kiwifruit bacterial canker [17, 18], and citrus bacterial canker [19, 20]. Zhang and Madhu Sekhar etal., found that the copper-based bactericides containing 1, 3, 4-thiadiazole fragments could prevent efficiently crop bacterial diseases attributed to cell membrane disruption [21], cytoderm disruption, and cell death of copper‐induced [17, 22].
In addition, RACS, another important commercial pesticide, exhibits excellent control efficacy for citrus bacterial canker [23]. While, the studies of RACS preventing rice bacterial leaf blight, and its antibacterial mechanism are rarely reported. Favaro etal., found that copper ion not only destroys cell membranes and cytoderm but also inhibits the formation of bacterial biofilm [24]. Ito and Dzięgielewska etal., found that resin acid not only inhibited the growth and biofilm formation of Streptococcus mutans [25] but also disturbed the biofilms’ biosynthesis of Staphylococcus aureus, Pseudomonas aeruginosa, and Candida albicans [26]. Resin acid amino alcohol derivatives based on the resin acid skeleton show excellent anti-virulence bioactivity against Xoo. In this work, we studied the bioactivity of RACS against Xoo in vitro and in vivo, and its potential anti-virulence mechanism was further interrogated.
Thank you for your kind suggestions again.
In the Results and Discussion part
Comments 8: The images not good.
Response 8: Thank you for your kind suggestions. We already had removed the black background from the picture labels A, B, and C, and enclosed them in brackets. Thank you for your kind suggestions again.
In the material and methods part
Comments 9: You have used the word “Firstly” in the beginning of different parts in the material and methods. Kindly use different words or just ignore it.
Response 9: Thank you for your kind suggestions. In the part of “3.5. Motility Assessment of RACS Against Xoo” and “3.6. The Analysis of Xoo-Extracellular Hydrolases Activity”, we already had removed the word “Firstly”, and replaced “Secondly” with “Then” in the revised manuscript. Thanks again.
Comments 10: In 3.6. part “Finally, the good or bad information was analyzed” This phrase must be changed or deleted from the text. However perhaps the software categories the results as “good” and “bad”; so you need to describe that.
Response 10: Thank you for your kind suggestions. The phrase of “Finally, the good or bad information was analyzed” was changed in the revised manuscript. In detail, the statement of “Finally, the good or bad information was analyzed” was removed and the statement of “Finally, the green filled circle for the good results, the red filled circle for the bad results, and the yellow filled circle for the general results.”. Thank you for your kind suggestions again.
The figures parts
Comments 11: Figure 10 as an example: this is your ligand: “. The Analysis of Pathogenicity of RACS on Rice Plant at 0 and 100 μg mL-1”: Kindly : 1) where the control? Where the description of “C”? “mL-1” need a correction (remove the full stop)
Response 11: Thank you for your kind suggestions. The treatment of “0 μg/mL” was control. The description of “C” and “mL-1” and Figure 10 was corrected in the revised manuscript. Thanks.
References
Comments 12: Nothing to concern
Response 12: Thank you for your kind suggestions.
General comment:
Comments 13: You must use a real enzyme activity test.
Response 13: Thank you for your kind suggestions. Your suggestion will be of great help to our research. We did not express clearly about this part. As shown in Figures 5A and 5B, we completed the experiment of real enzyme activity (extracellular amylase) using iodine solution staining. The results revealed that RACS significantly inhibited the activity of extracellular amylase and the inhibitory ratio was 33.75±7.21% at the dose of 100 μg mL−1. Moreover, we also tested the bioactivity of RACS against extracellular cellulase in Figure 1. The inhibition rate was 33.87 %. Thanks again.
Figure 1. The activity of Xoo-extracellular cellulase at the concentrations of control and 100 μg/mL.
Comments 14: The motility test is not representative, kindly give better images, or just change the experiment.
Response 14: Thank you for your kind suggestions. The bacterial motility is an important virulence factor in many pathogens [1-2]. It is an important index used to judge the virulence of bacteria [3]. So, the motility test is important to judge the virulence of Xoo. Thanks.
[1] Zarzecka, U.; Grinzato, A.; Kandiah, E.; Cysewski, D.; Berto, P.; Skorko-Glonek, J.; Zanotti, G.; Backert, S., Functional analysis and cryo-electron microscopy of Campylobacter jejuni serine protease HtrA. Gut Microbes. 2020, e1810532-2.
[2] Han, B.; Zheng, X. T.; Baruah, K.; Bossier, P., Sodium Ascorbate as a Quorum-Sensing Inhibitor Leads to Decreased Virulence in Vibrio campbellii. Front. Microbiol. 2020, 1054.
[3] Zhou, S. H.; Tu, X. T.; Pang, H. Y.; Hoare, R.; Monaghan, S. J.; Luo, J. J.; Jian, J. C., A T3SS Regulator Mutant of Vibrio alginolyticus Affects Antibiotic Susceptibilities and Provides Significant Protection to Danio rerio as a Live Attenuated Vaccine. Front. Cell. Infect. Microbiol. 2020, 183.
Comments 15: Kindly in most of the images there is no control. Kindly add some images to the control.
Response 15: Thank you for your kind suggestions. The treatment of “0 μg mL-1” was control in Figures 2-6 and 10. Additionally, we already had revised it in Figures 2-6 and 10 (See revised manuscript). Thanks.
Comments 16: You represent good images for the plants but you need the represent a correct killing to the microbes using your compound either by using a serial dilation method agar well diffusion experiment, etc.
Response 16: Thank you for your kind suggestions. Generally, we tested the EC50 value of RACS against microbes in Table 1 and it represented a correct killing or inhibiting to the microbes [1-3]. Moreover, in Figures 4 and 5, the bacterial motility and the activity of extracellular hydrolases was conducted using the semisolid medium including 0.5% agar. The results revealed that RACS displayed good inhibition activity for motility and extracellular hydrolases. Thanks.
[1] Yang, J.; Ye, H. J.; Xiang, H. M.; Zhou, X.; Wang, P. Y.; Liu, S. S.; Yang, B. X.; Yang, H. B.; Liu, L. W.; Yang, S., Photo‐stimuli smart supramolecular self‐assembly of azobenzene/β‐cyclodextrin inclusion complex for controlling plant bacterial diseases. Adv. Funct. Mater. 2023, 202303206.
[2] Liu, D. Y.; Song, R. J.; Wu, Z. X.; Xing, Z. F.; Hu, D. Y., Pyrido [1,2‑a] Pyrimidinone Mesoionic Compounds Containing Vanillin Moiety: Design, Synthesis, Antibacterial Activity, and Mechanism. J. Agric. Food Chem. 2022, 10443−10452.
[3] Ji, Q. T.; Hu, D. K.; Mu, X. F.; Tian, X. X.; Zhou, L.; Yao, S.; Wang, X. H.; Xiang, S. Z.; Ye, H. J.; Fan, L. J.; Wang, P. Y., Cucurbit[7]uril-Mediated Supramolecular Bactericidal Nanoparticles: Their Assembly Process, Controlled Release, and Safe Treatment of Intractable Plant Bacterial Diseases. Nano Lett. 2022, 4839-4847.
Thank you for your all kind suggestions of our manuscript. Your suggestion will be of great help to our research.

Reviewer 2 Report
Comments and Suggestions for Authors
The Manuscript ID: molecules-3146131 examined resin acid copper salt, as an interesting chemical pesticide, for controlling rice bacterial leaf blight by regulating bacterial virulence. It addresses an important and interesting subject. As a key commercial pesticide, resin acid copper salt (RACS) showed antibacterial mechanism and control efficacy of rice bacterial leaf blight in the study. The obtained data suggested that RACS manifested good curative activity (38.5%-45%) and sober anti-virulence function. While RACS blocked EPS’s biosynthesis by modulating the gumB gene expression and further decreasing the production of biofilm, it could also reduce bacterial virulence via regulating the motility and the activity of extracellular amylase. Through such modes of action, RACS could significantly minimize bacterial pathogenicity.
This manuscript is valid for publication after correcting grammar mistakes, flaws and typos in writing and presentation. Therefore, many sentences are somewhat confusing or disordered, to name a few:
i) Definite abbreviations should be spelled in full for the first time and thereafter may be abbreviated but not the opposite; e.g. Extracellular polysaccharide (EPS)… Resin acid copper salt (RACS)… Xanthomonas oryzae pv. oryzae (Xoo).
ii) While, the studies of RACS preventing rice bacterial leaf blight, and its antibacterial mechanism are rarely reported. …
iii) displays the anti-Xoo potency of RACS. The inhibitory rates were 100±2.31, 64.3±1.52, and 50.2±1.63 at the doses of 500, 100, and 50 μg mL−1, respectively.
iv) Table 2. The score of molinspiration bioactivity. This table is better replaced or phrased in a sentence instead.
v) microbial play an important role in nutrient.
Therefore, I would suggest accepting it after minor revision.
Comments on the Quality of English LanguageModerate editing of English language required.
Author Response
Comments 1: The Manuscript ID: molecules-3146131 examined resin acid copper salt, as an interesting chemical pesticide, for controlling rice bacterial leaf blight by regulating bacterial virulence. It addresses an important and interesting subject. As a key commercial pesticide, resin acid copper salt (RACS) showed antibacterial mechanism and control efficacy of rice bacterial leaf blight in the study. The obtained data suggested that RACS manifested good curative activity (38.5%-45%) and sober anti-virulence function. While RACS blocked EPS’s biosynthesis by modulating the gumB gene expression and further decreasing the production of biofilm, it could also reduce bacterial virulence via regulating the motility and the activity of extracellular amylase. Through such modes of action, RACS could significantly minimize bacterial pathogenicity.
Response 1: We thank the expert reviewer for the overall affirmative comment in regards of our manuscript.
Comments 2: This manuscript is valid for publication after correcting grammar mistakes, flaws and typos in writing and presentation. Therefore, many sentences are somewhat confusing or disordered, to name a few:
- i) Definite abbreviations should be spelled in full for the first time and thereafter may be abbreviated but not the opposite; e.g. Extracellular polysaccharide (EPS)…Resin acid copper salt (RACS)… Xanthomonas oryzae oryzae (Xoo).
Response 2: Thank you for your kind suggestions. We already had revised the abbreviations of EPS, RACS, and Xoo according to your suggestion in the revised manuscript. Thanks.
Comments 3: ii) While, the studies of RACS preventing rice bacterial leaf blight, and its antibacterial mechanism are rarely reported. …
Response 3: Thank you for your kind suggestions. We already had revised the sentence of “While, the studies of RACS preventing rice bacterial leaf blight and its antibacterial mechanism are rarely reported.” according to your suggestion in the revised manuscript. Thanks.
Comments 4: iii) displays the anti-Xoo potency of RACS. The inhibitory rates were 100±2.31, 64.3±1.52, and 50.2±1.63 at the doses of 500, 100, and 50 μg mL−1, respectively.
Response 4: Thank you for your kind suggestions. We already had revised the sentence of “displays the anti-Xoo potency of RACS. The inhibitory rates were 100±2.31, 64.3±1.52, and 50.2±1.63 at the doses of 500, 100, and 50 μg mL−1.” according to your suggestion in the revised manuscript. Thanks.
Comments 5: iv) Table 2. The score of molinspiration bioactivity. This table is better replaced or phrased in a sentence instead.
Response 5: Thank you for your kind suggestions. The Table 2 was replaced using a sentence in the revised manuscript. Thanks.
Comments 6: v) microbial play an important role in nutrient.
Response 6: Thank you for your kind suggestions. We already had revised it with your advice in the revised manuscript. Thanks.
Comments 7: Therefore, I would suggest accepting it after minor revision.
Response 7: We would like to express our deep appreciation to you for your affirmation of our research work.

Reviewer 3 Report
Comments and Suggestions for Authors
Dear Authors,
The article is interesting and contains valuable results. However, some elements need attention and correction before publication. Below are comments that could be considered by the authors for improving the manuscript.
- "was the major components of biofilm" – This phrase should be revised to "is the major component of biofilm." However, the structure of the biofilm can significantly vary depending on the bacterial species.
- Figures – Please revise the figure labels according to the journal's requirements. Remove the black background from the picture labels A, B, and C, and instead, enclose them in brackets.
- Section 2.5: The Inhibition Effect of Xoo-Extracellular Hydrolases – Why did the authors choose amylase as a marker? This enzyme does not seem as crucial in the infection of leaf tissue compared to cellulase or even pectinase. The choice of this enzyme should be clearly justified.
- The authors stated that RACS exhibited excellent bioactivity against Xoo in vitro. Could the authors provide information about similar substances and discuss why RACS is superior in this case?
- Section 2.8: The Control Effect Evaluation of RACS Containing 0.3% Orange Peel Essential Oil – Does the RACS water solution affect the water contact angle? Please add this information to the results section.
- "pathogenicity of RACS" – Please revise this phrase. Perhaps "influence of RACS on bacterial pathogenicity" would be more appropriate.
- Authors should focus more on the discussion of the results, as the current section mainly describes the findings with minimal references to the existing literature.
Author Response
Comments 1:
Dear Authors,
The article is interesting and contains valuable results. However, some elements need attention and correction before publication. Below are comments that could be considered by the authors for improving the manuscript.
Response 1: We thank the expert reviewer for the overall affirmative comment in regards of our manuscript.
Comments 2: 1. "was the major components of biofilm" – This phrase should be revised to "is the major component of biofilm." However, the structure of the biofilm can significantly vary depending on the bacterial species.
Response 2: Thank you for your kind suggestions. The “was the major components of biofilm” was revised to “is the major component of biofilm.” in the revised manuscript. We already had revised it with your advice in the revised manuscript. Thanks to the reviewer for the knowledgeable comments again. The structure of the biofilm can significantly vary depending on the bacterial species. The structure of Xoo-biofilm was displayed in Figure 1 based on our previous work [1]. Thanks.
Figure 1. The SEM of Xoo-biofilm.
[1] Chu, P. L.; Feng, Y. M.; Long, Z. Q.; Xiao, W. L.; Ji, J.; Zhou, X.; Qi, P. Y.; Zhang, T. H.; Zhang, H.; Liu, L. W.; Yang, S., Novel Benzothiazole Derivatives as Potential Anti-Quorum Sensing Agents for Managing Plant Bacterial Diseases: Synthesis, Antibacterial Activity Assessment, and SAR Study. J. Agric. Food Chem. 2023, 6525−6540.
Comments 3: 2. Figures – Please revise the figure labels according to the journal's requirements. Remove the black background from the picture labels A, B, and C, and instead, enclose them in brackets.
Response 3: Thank you for your kind suggestions. We already had removed the black background from the picture labels A, B, and C, and enclosed them in brackets. Thank you for your kind suggestions again.
Comments 4: 3. Section 2.5: The Inhibition Effect of Xoo-Extracellular Hydrolases – Why did the authors choose amylase as a marker? This enzyme does not seem as crucial in the infection of leaf tissue compared to cellulase or even pectinase. The choice of this enzyme should be clearly justified.
Response 4: Thanks to your knowledgeable comments. Cellulose and starch are the important loading-bearing skeletal component in plant cell walls. Bacteria often secreted cell wall-degrading enzymes, which contributed to bacterial virulence and included, e.g. cellulases, xylanases, polygalacturonases, and amylases. So, amylase is an important enzyme during the infection. Furthermore, we also tested the bioactivity of RACS against extracellular cellulase in Figure 2. The inhibition rate was 33.87 %. Thanks again.
Figure 2. The activity of Xoo-extracellular cellulase at the concentrations of control and 100 μg/mL.
Comments 5: 4. The authors stated that RACS exhibited excellent bioactivity against Xoo in vitro. Could the authors provide information about similar substances and discuss why RACS is superior in this case?
Response 5: Thank you for your kind suggestions. RACS exhibited excellent bioactivity against Xoo in vitro. The inhibition rate was 50.2±1.63 (%) at the dose of 50 μg/mL and the EC50 value was 50.0±4.43 (%). Dehydroabietic acid, a similar substances of resin acid, exhibited also good bioactivity against Xoo in vitro and the inhibition rate was 38.4±3.40 (%) at the dose of 50 μg/mL.[1] Moreover, thiodiazole copper, a commercial pesticide, exhibited also good bioactivity against Xoo in vitro. The inhibition rate was 46.8±2.2 (%) at the dose of 50 μg/mL and the EC50 value was 60.8±0.46 (%). [1,2] So, RACS exhibited better antibacterial activity than dehydroabietic acid and thiodiazole copper. Why RACS is superior in this case? It may be that the antibacterial activity was enhanced by the presence of copper ions and resin acid. Thanks again.
[1] Qi, P. Y.; Wang, N; Zhang, T. H.; Feng, Y. M.; Zhou, X.; Zeng, D.; Meng, J.; Liu, L. W.; Jin, L. H.; Yang, S., Anti-Virulence Strategy of Novel Dehydroabietic Acid Derivatives: Design, Synthesis, and Antibacterial Evaluation. Int. J. Mol. Sci. 2023, 2897.
[2] Chu, P. L.; Feng, Y. M.; Long, Z. Q.; Xiao, Wan-Lin.; Jin Ji, Xiang Zhou, Pu-Ying Qi, Tai-Hong Zhang, Heng Zhang, Li-Wei Liu, Song Yang., Novel Benzothiazole Derivatives as Potential Anti-Quorum Sensing Agents for Managing Plant Bacterial Diseases: Synthesis, Antibacterial Activity Assessment, and SAR Study. J. Agric. Food Chem. 2023, 6525−6540.
Comments 6: 5. Section 2.8: The Control Effect Evaluation of RACS Containing 0.3% Orange Peel Essential Oil – Does the RACS water solution affect the water contact angle? Please add this information to the results section.
Response 6: Thank you for your kind suggestions. As shown in Figure 3, The contact angle of RACS (20%, m/v, Emulsion in Water) was 59°. Thanks.
Figure 3. The contact angle of RACS (20%, m/v, Emulsion in Water).
Comments 7: 6. "pathogenicity of RACS" – Please revise this phrase. Perhaps "influence of RACS on bacterial pathogenicity" would be more appropriate.
Response 7: Thank you for your kind suggestions. The phrase of “pathogenicity of RACS” is replaced in a sentence of “influence of RACS on bacterial pathogenicity”. Thanks.
Comments 8: 7. Authors should focus more on the discussion of the results, as the current section mainly describes the findings with minimal references to the existing literature.
Response 8: Thank you for your kind suggestions. We already had revised the discussions with your suggestion. (1) The discussion of the result of “2.3. The Analysis of Xoo-Biofilm Formation Using Acridine Orange Staining” was corrected in the revised manuscript. In detain, “These outcomes were analogous to crystal violet staining. Based on these findings, RACS could significantly suppress bacterial virulence by regulating gumB gene and biofilm’s biosynthesis. Thereby, RACS might be regarded as a virulence inhibitor for controlling plant disease.” (2) The discussion of the result of “2.4. The Inhibited Effects on Xoo-Motility” was corrected in the revised manuscript. In detai, “Therefore, Xoo-motility was reduced markedly under the influence of RACS. It may be that the antibacterial activity was enhanced by the presence of copper ions and resin acid. Copper ions and resin acid could disturb the formation of biofilm. Meanwhile, the reduction of Xoo-motility might result in the decline of bacterial virulence, which ena-bled bacteria not to adapt the environmental changes, decreased effectively bacterial infections, and obviously reduced the morbidity of plant diseases. So, RACS might be an inhibitor of Xoo-motility.” Thanks.
Thank you for your all kind suggestions of our manuscript. Your suggestion will be of great help to our research.

Round 2
Reviewer 1 Report
Comments and Suggestions for Authors
Dear authors
Happy day
Many thanks for considering all the addressed points. The paper now is fine.
With my pleasure

Author Response
Comment:
Dear authors
Happy day
Many thanks for considering all the addressed points. The paper now is fine.
With my pleasure
Response: We thank for your affirmative comment in the revised manuscript. Thanks again.